# No Harmful Effect of Endovascular Treatment before Decompressive Surgery—Implications for Handling Patients with Space-Occupying Brain Infarction

**DOI:** 10.3390/jcm13030918

**Published:** 2024-02-05

**Authors:** Johann Otto Pelz, Simone Engelmann, Cordula Scherlach, Peggy Bungert-Kahl, Alhuda Dabbagh, Dirk Lindner, Dominik Michalski

**Affiliations:** 1Department of Neurology, University Hospital Leipzig, 04103 Leipzig, Germany; 2Institute of Neuroradiology, University Hospital Leipzig, 04103 Leipzig, Germany; 3Neurological Rehabilitation Center Leipzig, 04828 Bennewitz, Germany; 4Department of Neurosurgery, University Hospital Leipzig, 04103 Leipzig, Germany

**Keywords:** stroke, space-occupying brain infarction, malignant middle cerebral artery infarction, mechanical thrombectomy, decompressive hemicraniectomy

## Abstract

This study explored short- and mid-term functional outcomes in patients undergoing decompressive hemicraniectomy (DHC) due to space-occupying cerebral infarction and asked whether there is a potentially harmful effect of a priorly performed endovascular treatment (EVT). Medical records were screened for patients requiring DHC due to space-occupying cerebral infarction between January 2016 and July 2021. Functional outcomes at hospital discharge and at 3 months were assessed by the modified Rankin Scale (mRS). Out of 65 patients with DHC, 39 underwent EVT before DHC. Both groups, i.e., EVT + DHC and DHC alone, had similar volumes (280 ± 90 mL vs. 269 ± 73 mL, *t*-test, *p* = 0.633) and proportions of edema and infarction (22.1 ± 6.5% vs. 22.1 ± 6.1%, *t*-test, *p* = 0.989) before the surgical intervention. Patients undergoing EVT + DHC tended to have a better functional outcome at hospital discharge compared to DHC alone (mRS 4.8 ± 0.8 vs. 5.2 ± 0.7, Mann–Whitney-U, *p* = 0.061), while the functional outcome after 3 months was similar (mRS 4.6 ± 1.1 vs. 4.8 ± 0.9, Mann–Whitney-U, *p* = 0.352). In patients initially presenting with a relevant infarct demarcation (Alberta Stroke Program Early CT Score ≤ 5), the outcome was similar at hospital discharge and after 3 months between patients with EVT + DHC and DHC alone. This study provided no evidence for a harmful effect of EVT before DHC in patients with space-occupying brain infarction.

## 1. Introduction

Space-occupying brain infarction, also termed as malignant middle cerebral artery infarction, represents the maximum version of ischemic stroke. This condition is typically caused by proximal and thus large vessel occlusion (LVO) and is characterized by mortality rates of up to 80% or long-lasting disability in most survivors [1,2]. Currently, treatment strategies focus on early decompressive hemicraniectomy (DHC) to provide additional space for the developing edema to prevent cerebral herniation [3,4]. 

In 2015, the introduction of endovascular treatment (EVT) marked a milestone in treating acute ischemic stroke due to LVO [5,6]. Whereas first studies mainly focused on patients with no or only small infarct demarcation, recent randomized controlled trials provided evidence for beneficial effects of EVT also in the situation of an already existing relevant infarct demarcation [7,8,9]. Along with an increasing implementation of EVT in acute stroke management, data on individual courses emerging from the real-world setting indicated a decreased rate of space-occupying brain infarction [10,11,12], in conjunction with a reduced rate of DHC [13,14,15]. 

However, despite the progress in acute stroke treatment, the need for DHC still exists and is typically seen in approximately 2% of patients in clinical studies [13,15]. In patients exhibiting a relevant brain infarction before EVT, even a rate of up to 7.4% for DHC was observed [8]. DHC after ischemic stroke harbors a relevant risk of complications: hemorrhagic complications such as ipsi- or contralateral hematoma occurred in 20.7% of patients, while hemorrhagic transformation of the ischemic lesion was observed in 23.7% of cases undergoing DHC [16]. Noteworthy, these data emerged from the time before the wide-spread implementation of EVT in acute stroke management. On the other hand, hemorrhagic transformation is also a typical complication in ischemic stroke independently of EVT [17,18]. However, abrupt reperfusion of LVO might aggravate the hemorrhagic transformation or even result in parenchymal hemorrhage after DHC, which might ultimately result in a poorer functional outcome of these patients. 

To explore a potential harmful effect of recanalization approaches before decompressive surgery, this study investigated the short- and mid-term functional outcomes in patients requiring DHC due to space-occupying brain infarction after EVT compared to those without endovascular recanalization attempts. To consider a situation regularly occurring in clinical practice, a subgroup analysis focused on EVT with relevant infarct demarcation upon admission, i.e., an Alberta Stroke Program Early CT Score (ASPECTS) ≤ 5.

## 2. Materials and Methods

This retrospective, non-interventional, explorative study was performed according to the ethical standards laid down in the 1964 Declaration of Helsinki and its later amendment. The study was approved by the local ethics committee of the Medical Faculty at the University of Leipzig (reference number 179/21-ek).

Hospital-based medical records of all stroke patients treated in the Department of Neurology at the University of Leipzig between January 2016 and July 2021 were screened. Patients were included if they required DHC due to a space-occupying brain infarction, which was defined according to the inclusion criteria of the DESTINY trials [19,20]. Basically, these criteria include an infarction of at least two-thirds of the middle cerebral artery territory and a decreased level of consciousness. An additional imaging criterion was applied if patients underwent EVT in general anesthesia and were still sedated at the time of infarct reevaluation. In detail, these patients had to present a relevant compression of the ipsilateral ventricular systems or a mid-line shift towards the contralateral hemisphere on computed tomography (CT) after EVT. As a further inclusion criterion, patients had to follow an entire treatment course, which included, according to local standards and closely following the DESTINY trials [19,20], DHC and a bundle of actions with, for instance, continuous intracerebral pressure monitoring, deep anesthesia, normothermia, and, if necessary, osmotherapy.

Closely adhering to the milestone studies (e.g., [5,6]), EVT was performed in patients presenting with LVO and relevant clinical symptoms. Due to individual characteristics and existing studies that also showed efficacy in a time window of up to 16 and 24 h [21,22], a few patients were treated in an extended time window based on imaging criteria including CT perfusion. Concerning imaging criteria, patients typically had to present only marginal demarcation in terms of an ASPECTS of >5 in the initial non-contrast CT [23]. In case of an ASPECTS ≤ 5, the decision to perform EVT was made in consent between the senior neurologist and senior neuroradiologist based on cerebral perfusion imaging. 

In addition to demographic data and clinical characteristics during the hospital stay, information regarding stroke treatment (intravenous thrombolysis, mechanical thrombectomy and associated parameters) were extracted from medical records. Moreover, the initial non-contrast CT was re-evaluated by a neuroradiologist who was blinded to clinical and treatment data of the patients to re-assess the initial ASPECTS and the expanded Thrombolysis in Cerebral Infarction (eTICI) score after EVT [24]. 

Using the software MRIcron (v1.0.20190902, University of South Carolina Center: McCausland Center for Brain Imaging, https://www.nitrc.org/projects/mricron; accessed on 25 May 2022), the individual volume of hypodense infarct zone with associated edema and the proportion between the hypodense infarct zone with associated edema and the supra- and infratentoriell brain tissue (whole brain) were calculated by manually marking these areas on the last CT conducted prior to DHC. This procedure was performed by an investigator who was blinded to clinical and other imaging or peri-procedural data. 

The functional outcome was assessed by the modified Rankin Scale (mRS) at hospital discharge and after 3 months (±14 days).

The software SPSS (versions 27.0 and 29.0, IBM Corp., Armonk, NY, USA) was used for statistical calculations. After descriptive analyses, statistical significance between groups were assessed by chi-square test for categorical variables and by *t*-test or Mann–Whitney U test, depending on whether the respective parameters were normally distributed or not. Thereby, a *p*-value < 0.05 was defined as statistically significant. 

## 3. Results

Sixty-five patients were identified as undergoing DHC due to space-occupying brain infarction and fulfilling the study inclusion criteria. Of these patients, 39 (60%) had prior endovascular treatment. mRS at hospital discharge was available from 65 patients and after 3 months from 63 patients, as two patients were lost to follow-up after hospital discharge. Demographic data of the overall sample and the distributions to treatment groups are shown in Table 1. Both groups, i.e., patients treated with EVT and DHC and patients without EVT, did not differ regarding age and sex. 

In the overall study population, patients who underwent EVT before DHC exhibited a significantly higher ASPECTS at hospital admission compared to cases with DHC alone (6.4 ± 2.6 vs. 2.7 ± 1.6, Mann–Whitney U test, *p* < 0.001, Table 1). However, no significant differences between patients with and without EVT before DHC were seen regarding volume (280 ± 90 mL vs. 269 ± 73 mL, *t*-test, *p* = 0.633) and proportion of cerebral edema and infarction, referring to the whole brain volume (22.1 ± 6.5% vs. 22.1 ± 6.1%, *t*-test, *p* = 0.989), before the surgical intervention (Table 1). There was a trend towards a better functional outcome at discharge for patients undergoing sequential EVT and DHC compared to cases with DHC alone (mRS 4.8 ± 0.8 vs. 5.2 ± 0.7, Mann–Whitney U test, *p* = 0.061; Figure 1). After 3 months, functional outcome between both groups was similar (Figure 1). When dichotomizing patients for a good outcome (mRS 0-3) and poor outcome (mRS 4-6) at 3 months, 5 out of 38 patients achieved good outcomes after sequential EVT and DHC, and none out of 25 patients achieved good outcome after surgical treatment alone (chi-square test, *p* = 0.059).

To cover a regularly occurring situation in clinical practice, a subgroup analysis focused on patients exhibiting a relevant infarct demarcation (ASPECTS ≤ 5) at the time of hospital admission. Although the initial ASPECTS in this subgroup significantly differed by about one point between patients with sequential EVT and DHC and those undergoing DHC alone (3.9 ± 1.5 vs. 2.7 ± 1.6, Mann–Whitney U test, *p* = 0.021, Table 2), both groups did not differ concerning the volume and proportion of cerebral edema and infarction on the last available cerebral imaging before surgery (Table 2). Functional outcome at discharge and after 3 months was similar between both groups (Figure 2).

To explore the potential impact of recanalization referring to the functional outcome in patients suffering from stroke due to LVO in the overall sample, outcome was dichotomized for good (mRS 0-3) and poor (mRS 4-6), and respective proportions were analyzed depending on degree of recanalization. Among patients with relevant recanalization (eTICI 2b or better), 3 out of 24 patients (12.5%) reached a good functional outcome after 3 months, while only 2 out of 39 patients (5.1%) with recanalization lower than eTICI 2b or not attempted recanalization had a good functional outcome, which, however, did not reach statistical significance (chi-square test, *p* = 0.360).

## 4. Discussion

While investigating the functional outcome in patients requiring DHC due to space-occupying brain infarction, this study asked for a potential harmful effect of a priorly performed EVT. As the main finding, there was no evidence for a harmful effect of EVT before DHC regarding the short- and mid-term functional outcome in patients with space-occupying brain infarction. Instead, a trend was seen towards an even improved outcome after sequential EVT and DHC compared to DHC alone. 

Our findings are in line with a very recent study reporting no increased rate of early surgical complications and a similar functional outcome after about 5 and 14 months of patients with EVT and DHC compared to patients undergoing DHC alone [25]. Regarding complications directly related to the surgical procedure, an earlier study described a comparable intraoperative blood loss, an equal duration of surgery, and similar rates of intra- or extracranial bleeding complications were found when comparing patients who underwent DHC and patients with sequential endovascular and surgical treatment [26].

In addition, we also observed no harmful effect of EVT in patients initially exhibiting a relevant infarct demarcation (ASPECTS ≤ 5). This observation is relevant as recent randomized controlled trials have demonstrated beneficial effects of EVT in patients with larger infarct demarcation, i.e., ASPECTS ≤ 5 [7,8,9]. It can therefore be assumed that in the future, more patients will be treated with EVT, resulting in a relevant infarct demarcation even though blood flow is sufficiently restored, and thus need decision making regarding DHC. Even for this situation, the results from the subgroup analyses of the present study do not indicate a harmful effect of surgery after endovascular treatment.

Regarding the amount of cerebral edema and infarction, a recent study has found that a volume of >258 mL before DHC was associated with worse outcomes, defined as mRS 4 to 6 [27]. Considering a mean edema and infarct volume of 276 mL with no relevant differences between groups before DHC in the present study, it is noticeable that sequential EVT and DHC was found to provide at least a trend towards an improved outcome compared to DHC alone.

So far, efforts were made to early identify patients with an increased risk of developing a space-occupying cerebral infarction after LVO and EVT. An imaging-based study identified the occlusion site (i.e., the internal carotid artery) and a worse collateral status as the main factors associated with edema development despite subsequent successful recanalization [28]. In a multi-parametric investigation, more severe clinical symptoms at baseline, an initial ASPECTS ≤ 8, the thrombus burden (i.e., the number of occluded segments), and an unsuccessful recanalization were found to be associated with the need for DHC [29]. However, identifying early assessable risk factors for developing a space-occupying infarction in patients with LVO might carry the inherent risk of considering any recanalization attempt as futile or even harmful because of the fear of reperfusion injury and the inevitable need for DHC. As indicated by the findings of this study, this fear appears not justified since an EVT before DHC was not associated with detrimental effects regarding the functional outcome. 

Since the sequence of EVT and DHC seems to be safe, clinical studies searching for neuroprotectants in patients undergoing EVT with a lower ASPECTS appear reasonable, regardless of a potentially emerging need for subsequent DHC. In this sense, a promising candidate is nerinetide that is supposed to alleviate excitotoxic effects of ischemia when administered in the context of EVT [30].

This study has some limitations: First, the number of patients is relatively small and thus prevents a generalization of findings and subgroup analyses, especially concerning intravenous thrombolysis, pre-medication, sex-related effects, and the time window from symptom onset and performed EVT or DHC. However, even if patients differed regarding the initial ASPECTS or the reperfusion state, the proportion of the cerebral edema and infarction referring to the whole brain volume was identical on the last cerebral CT before decompressive surgery, alleviating the importance of these parameters in the setting investigated here. As our study included only patients with DHC, the initial ASPECTS can thus be seen as a snapshot, while the growth of the ischemic lesion with an ultimately space-occupying effect depends on a variety of conditions, e.g., the collateral state. Consequently, some patients with LVO typically provide a slow evolution of the ischemic core and would thus benefit from EVT in an extended time window, also called the ‘late window paradox’ [31]. These patients would probably not develop a space-occupying cerebral edema and, thus, were not examined in this study. Second, the retrospective study design only allows descriptive analyses and the calculation of statistical relationships, while for explorations of causal relationships a prospective, preferably randomized design, would be necessary. However, randomization concerning endovascular and conservative treatment in patients with LVO appears unfeasible for ethical reasons, as previous studies have clearly shown beneficial effects for EVT (e.g., [5,6,8]). Third, when focusing on functional outcome of patients with space-occupying cerebral infarction, further treatment after DHC may considerably vary [32]. For example, even subtle factors like the fluid balance during the acute phase may have an impact on the functional outcome [33]. 

## 5. Conclusions

This study provided insights into the clinical course, i.e., the short- and mid-term functional outcome, of patients undergoing DHC due to space-occupying brain infarction and prior EVT. Based on the observed course of patients without an EVT before DHC, no evidence was found for a harmful effect of EVT before the surgical procedure. Together with the trend towards an improved outcome after sequential EVT and DHC, these findings might reduce concerns during decision making in patients with space-occupying brain infarction. Furthermore, this study supports the perspective of an overall beneficial effect of recanalizing strategies in ischemic stroke due to large vessel occlusion.

## Figures and Tables

**Figure 1 jcm-13-00918-f001:**
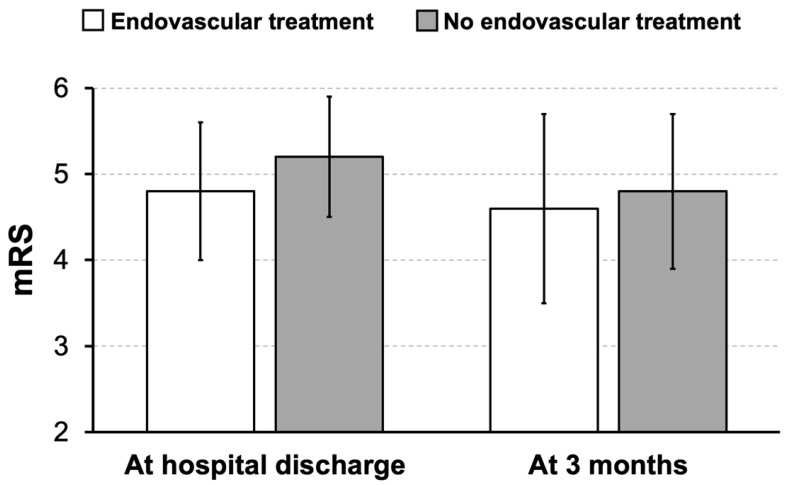
Overall study sample: functional outcome in patients undergoing sequential endovascular treatment and decompressive hemicraniectomy compared with patients undergoing surgical treatment alone. Bars represent mean values and added lines standard deviation of mean. mRS: modified Rankin scale.

**Figure 2 jcm-13-00918-f002:**
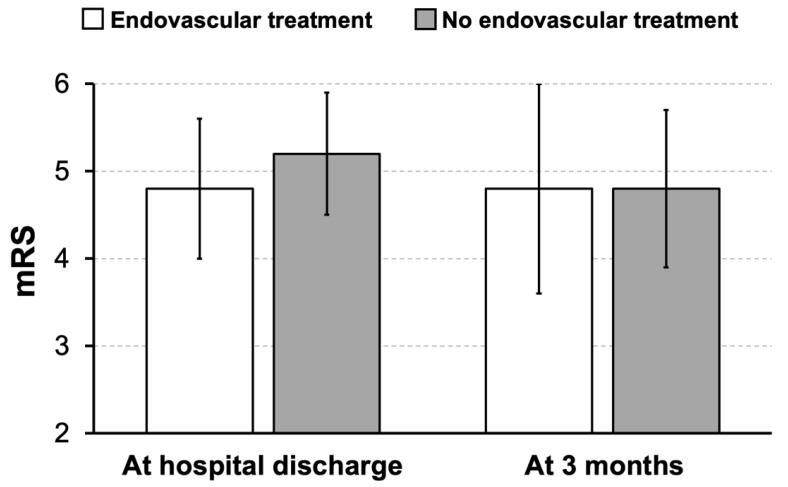
Subgroup analysis of patients presenting with relevant infarct demarcation (ASPECTS ≤ 5). Bars represent mean values and added lines standard deviation of mean. There were not differences in the functional outcome between both groups at discharge nor at 3 months. mRS: modified Rankin scale.

**Table 1 jcm-13-00918-t001:** Demographic, imaging- and recanalization-related characteristics in the overall sample and depending on treatment.

	Overall Sample	Endovascular Treatment	No Endovascular Treatment	*p*-Value
n	65	39	26	
Age in years (M ± SD)	59 ± 11	58 ± 12	61 ± 9	0.088 ^+^
Sex (% female) (f/m)	30.8 (20/45)	28.2 (11/28)	34.6 (9/17)	0.583 ^#^
ASPECTS of the first non-contrast CT (M ± SD)	5.0 ± 2.9	6.4 ± 2.6	2.7 ± 1.6	<0.001 *
Intravenous thrombolysis (% (n/n))	32.3 (21/65)	41.0 (16/39)	17.9 (5/26)	0.066 ^#^
eTICI (n)		0 = 3	-	-
		1 = 6		
		2a = 5		
		2b = 10		
		2c = 2		
		3 = 13		
Volume (in mL, M ± SD) and proportion (%, M ± SD) of edema/infarction referring to whole brain volume on CT prior to hemicraniectomy ^1^	276 ± 83 22.1 ± 6.3	280 ± 90 22.1 ± 6.5	269 ± 73 22.1 ± 6.1	0.633 ^+^ 0.989 ^+^

Abbreviations: CT: computed tomography, ASPECTS: Alberta Stroke Program Early CT Score, eTICI: expanded Thrombolysis in Cerebral Infarction, M: mean value, SD: standard deviation, ^1^ Analysis of relative volume was based on 60 out of 65 patients, ^+^: *t*-test, *: Mann–Whitney U test, ^#^: chi-square test.

**Table 2 jcm-13-00918-t002:** Demographic, imaging- and recanalization-related characteristics in patients exhibiting a relevant infarct demarcation (ASPECTS ≤ 5) at hospital admission depending on treatment.

	Overall Sample	Endovascular Treatment	No Endovascular Treatment	*p*-Value
n	41	16	25	
Age in years (M ± SD)	60 ± 10	57 ± 11	61 ± 9	0.150 ^+^
Sex (% female) (f/m)	29.3 (12/29)	18.8 (3/13)	36.0 (9/16)	0.236 ^#^
ASPECTS of the first non-contrast CT (M ± SD)	3.2 ± 1.6	3.9 ± 1.5	2.7 ± 1.6	0.021 *
Patients with an ASPECTS between 3 and 5 (% (n/n)) on the first non-contrast CT	68.3 (28/41)	87.5 (14/16)	60.0 (15/25)	0.059 ^#^
Intravenous thrombolysis (% (n/n))	19.5 (8/41)	18.8 (3/16)	20.0 (5/25)	0.921 ^#^
eTICI (n)		0 = 1	-	-
		1 = 2		
		2a = 2		
		2b = 3		
		2c = 2		
		3 = 6		
Volume (in mL, M ± SD) and proportion (%, M ± SD) of edema/infarction referring to whole brain volume on CT prior to hemicraniectomy ^1^	283 ± 89 22.8 ± 6.6	297 ± 110 23.4 ± 7.6	273 ± 72 22.4 ± 6.0	0.421 ^+^ 0.655 ^+^

Abbreviations: CT: computed tomography, ASPECTS: Alberta Stroke Program Early CT Score, eTICI: expanded Thrombolysis in Cerebral Infarction, M: mean value, SD: standard deviation, ^1^ Analysis of relative volume was based on reduced patient number, ^+^: *t*-test, *: Mann–Whitney U test, ^#^: chi-square test.

## Data Availability

All data generated or analyzed during this study are included in this published article and are available from the corresponding author upon reasonable request.

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
