# Peer review of "No Harmful Effect of Endovascular Treatment before Decompressive Surgery—Implications for Handling Patients with Space-Occupying Brain Infarction"

_jcm, 2024, doi:10.3390/jcm13030918_

Round 1
Reviewer 1 Report
Comments and Suggestions for Authors
This study aims to investigate short-and mid- term functional outcomes in patients undergoing DC due to space occupying brain infarcts and investigate as well a potentially harmful effect of a priorly performed EVT.
There are several issues regarding size of the sample (may be a two center study?) and methodology.
1. The abstract is not adequately informative
2. In the Methods section:
Regarding the justification for the use of the Mann-Whitney U test, parametric testing can be used when dealing with small samples. The MWU test is usually chosen when the normality hypothesis for parametric testing is violated.
The authors should strike the whole sentence.
3. In the result section :
There is a lack of structure and the result presentation is not clear. The authors do not specify what test is being used to examine each hypothesis and the presented results are lacking.
When presenting the results of hypothesis the authors should mention a) what test is being used b) if the test's assumptions are being met c) the test's results (e.g mean difference , p value) both in the result section of the manuscript and the accompanying table.
e.g
- "In the overall study population, patients who underwent endovascular treatment before decompressive hemicraniectomy exhibited a significantly higher ASPECTS at hospital admission and received intravenous thrombolysis numerically more often than patients without endovascular treatment."
This sentence should be accompanied by a measure of difference and a statistical value for the examined variable (in this case the Aspects score)
- "2 out of 16 patients achieved good outcomes after sequential endovascular treatment and decompressive hemicraniectomy and none of 24 patients achieved good outcome after surgical treatment only (p = 0.154)."
Again the authors present a single p-value without ever addressing which test is being used, the actual test's result or what constitutes a good or bad outcome.
- In at least 3 instances the author report " a numerical difference" without actually mentioning said difference in text.
no comments
Author Response
Please see the attachment.
Dear Dr. …,
Thank you very much for considering our manuscript for possible publication in Journal of Clinical Medicine. Here, we want to submit a manuscript version that has been revised with reference to the reviewers’ valuable comments.
Please find below a detailed point-to-point reply and a description of changes made regarding the given comments. Throughout the manuscript, revised sentences are marked in blue font.
Please do not hesitate to contact us for any further questions or considerations.
Yours sincerely,
Johann Pelz and Dominik Michalski
Reviewer 1
- The abstract is not adequately informative.
Reply: The abstract was rewritten and in particular the data and statistical analyses underlying the results were added:
This study explored short- and mid-term functional outcomes in patients undergoing decompressive surgery (DHC) due to space-occupying cerebral infarction and asked for a potentially harmful effect of a priorly performed endovascular treatment (EVT).
Medical records were screened for patients requiring DHC due to space-occupying cerebral infarction between 01/2016 and 07/2021. Functional outcomes at hospital discharge and at 3 months were assessed by the modified Rankin Scale (mRS).
Out of 65 patients with DHC, 39 underwent EVT before DHC. Both groups, i.e., EVT+DHC and DHC alone, had similar volumes (280±90 ml vs. 269±73 ml, t-test, p=0.633) and proportions of edema and infarction (22.1±6.5% vs. 22.1±6.1%, t-test, p=0.989) before the surgical intervention. Patients undergoing EVT+DHC tended to have a better functional outcome at hospital discharge compared to DHC alone (mRS 4.8±0.8 vs. 5.2±0.7, Mann-Whitney-U, p=0.061), while the functional outcome after 3 months was similar (mRS 4.6±1.1 vs. 4.8±0.9, Mann-Whitney-U, p=0.352). In patients initially presenting with a relevant infarct demarcation (Alberta Stroke Program Early CT Score ≤5), the outcome was similar at hospital discharge and after 3 months between patients with EVT+DHC and DHC alone.
This study provided no evidence for a harmful effect of EVT before DHC in patients with space-occupying brain infarction.
- In the Methods section: Regarding the justification for the use of the Mann-Whitney U test, parametric testing can be used when dealing with small samples. The MWU test is usually chosen when the normality hypothesis for parametric testing is violated. The authors should strike the whole sentence.
Reply: We agree with the reviewer and chose a parametric (t-test) or non-parametric (Mann-Whitney-U test depending on whether the data was normally distributed or not. We clarified this in the methods section:
After descriptive analyses, statistical significance between groups were assessed by chi-square test for categorical variables and by t-test or Mann-Whitney U test, depending on whether the respective parameters were normally distributed or not.
- In the result section: There is a lack of structure and the result presentation is not clear. The authors do not specify what test is being used to examine each hypothesis and the presented results are lacking.
When presenting the results of hypothesis the authors should mention a) what test is being used b) if the test's assumptions are being met c) the test's results (e.g mean difference , p value) both in the result section of the manuscript and the accompanying table.
In at least 3 instances the author report " a numerical difference" without actually mentioning said difference in text.
Reply: The results section was revised according to the suggestions of the reviewer. The structure is clearer, since we now report the results of the overall sample and then of a relevant subgroup (patients with an initial ASPECTS <6). The respective values and statistical tests were added. The phrase “numerical difference” was removed from the manuscript:
“3. Results
Sixty-five patients were identified as undergoing DHC due to space-occupying brain infarction and fulfilling the study inclusion criteria. Of these patients, 39 (60%) had prior endovascular treatment. mRS at hospital discharge was available from 65 patients and after 3 months from 63 patients, as two patients were lost to follow-up after hospital discharge. Demographic data of the overall sample and the distributions to treatment groups are shown in Table 1. Both groups, i.e., patients treated with EVT and DHC and patients without endovascular treatment, did not differ regarding age and sex.
In the overall study population, patients who underwent EVT before DHC exhibited a significantly higher ASPECTS at hospital admission compared to cases with DHC alone (6.4 ± 2.6 vs. 2.7 ± 1.6, Mann-Whitney-U test: p < 0.001). However, no significant differences between patients with and without EVT before DHC were seen regarding volume (280 ± 90 ml vs. 269 ± 73 ml, t-test: p = 0.633) and proportion of cerebral edema and infarction, referring to the whole brain volume (22.1 ± 6.5% vs. 22.1 ± 6.1%, t-test: p = 0.989), before the surgical intervention. There was a trend towards a better functional outcome at discharge for patients undergoing sequential EVT and DHC compared to cases with DHC alone (mRS 4.8 ± 0.8 vs. 5.2 ± 0.7, Mann-Whitney-U test: p = 0.061; Figure 1). After 3 months, functional outcome between both groups was similar (Figure 1). When dichotomizing patients for a good outcome (mRS 0-3) and poor outcome (mRS 4-6) at 3 months, 5 out of 38 patients achieved good outcomes after sequential EVT and DHC, and none of 25 patients achieved good outcome after surgical treatment alone (chi-square test: p = 0.059).
To cover a regularly occurring situation in clinical practice, a subgroup analysis focused on patients exhibiting a relevant infarct demarcation (ASPECTS ≤ 5) at the time of hospital admission. Although the initial ASPECTS in this subgroup significantly differed by about one point between patients with sequential EVT and DHC and those undergoing DHC alone (3.9 ± 1.5 vs. 2.7 ± 1.6, Mann-Whitney-U test: p = 0.021), both groups did not differ concerning the volume and proportion of cerebral edema and infarction on the last available cerebral imaging before surgery (Table 2). Functional outcome at discharge and after 3 months was similar between both groups (Figure 2).
To explore the potential impact of recanalization referring to the functional outcome in patients suffering from stroke due to LVO in the overall sample, outcome was dichotomized for good (mRS 0-3) and poor (mRS 4-6), and respective proportions were analyzed depending on degree of recanalization. Among patients with relevant recanalization (eTICI 2b or better), 3 out of 24 patients (12.5%) reached a good functional outcome after 3 months, while only 2 out of 39 patients (5.1%) with recanalization lower than eTICI 2b or not attempted recanalization had a good functional outcome, which, however, did not reach statistical significance (chi-square test, p = 0.360).“
Reviewer 2 Report
Comments and Suggestions for Authors
The presented article is topical, reflecting the necessity to use in patients with ischemic stroke due to the large vessel occlusion not only pathogenetic therapy (decompressive hemicraniectomy (DC), correction of ionic imbalance, etc.), but also etiotropic, which in this case is the primary performed endovascular treatment (EVT). I assume that if the authors will have an opportunity to continue their research work prospectively, it is worth studying in more detail the dynamics of prooxidant and antioxidant systems in this pathology (for example, to evaluate and correlate the accumulation of ionized oxygen in arterial blood, activity of superoxide dismutase, catalase, changes in concentrations of copper, iron, selenium, etc.) in order to assess ischemic and reperfusion changes in tissues and their prevention. Perhaps this will help the authors to confirm the need statistically reliably for EVT before DC in all patients with ischemic stroke due to the large vessel occlusion.
Regards, reviewer.
Author Response
Please see the attachment.
Dear Dr. …,
Thank you very much for considering our manuscript for possible publication in Journal of Clinical Medicine. Here, we want to submit a manuscript version that has been revised with reference to the reviewer’s valuable comments.
Please find below a detailed point-to-point reply and a description of changes made regarding the given comments. Throughout the manuscript, revised sentences are marked in blue font.
Please do not hesitate to contact us for any further questions or considerations.
Yours sincerely,
Johann Pelz and Dominik Michalski
Reviewer 2
The presented article is topical, reflecting the necessity to use in patients with ischemic stroke due to the large vessel occlusion not only pathogenetic therapy (decompressive hemicraniectomy (DC), correction of ionic imbalance, etc.), but also etiotropic, which in this case is the primary performed endovascular treatment (EVT). I assume that if the authors will have an opportunity to continue their research work prospectively, it is worth studying in more detail the dynamics of prooxidant and antioxidant systems in this pathology (for example, to evaluate and correlate the accumulation of ionized oxygen in arterial blood, activity of superoxide dismutase, catalase, changes in concentrations of copper, iron, selenium, etc.) in order to assess ischemic and reperfusion changes in tissues and their prevention. Perhaps this will help the authors to confirm the need statistically reliably for EVT before DC in all patients with ischemic stroke due to the large vessel occlusion.
Reply: We thank the reviewer for his comment. Recently, the interest in neuroprotectant drugs experienced a revival. The situation in patients with large vessel occlusion undergoing endovascular treatment is similar to preclinical studies where the occluded vessel is re-opened at a defined time point. Amongst others, nerinetide is such a prominsing candidate when investigating neuroprotectants in acute stroke due to large vessel occlusion. After positive signals in subgroup analyses of the ESCAPE-Na1, nerinetide was administered during endovascular treatment in patients with large vessel occlusion but without prior thrombolysis (ESCAPE NEXT). However, so far, the results of ESCAPE NEXT have not been published. We added this considerations to the discussion.:
Since the sequence of EVT and DHC seems to be safe, clinical studies searching for neuroprotectants in patients undergoing EVT with a lower ASPECTS appears reasonable, regardless of a potentially emerging need for subsequent DHC. In this sense, a promising candidate is nerinetide that is supposed to alleviate excitotoxic effects of ischemia when administered in the context with EVT [30].
Reviewer 3 Report
Comments and Suggestions for Authors
The Authors report a study on 65 patients retrospectively recruited among pt. qith decompressive surgery due to ischemic stroke, 39 with endovascular treatment (EVT). Comparing the groups with and without EVT they found higher ASPECT in the EVT group, no statistically significant difference in mRS at discharge, while there was a difference in dichotomized mRS at 3 months.
The topic is of interest, yet the paper deserves some improvements:
1. it would be important to know the "vessel status" (in terms ov occluded vessel and collateral score) in the two groups
2. the higher ASPECT lin EVT can be the explanation of better prognosis at 3 months;
3. as the two groups differs terms of ASPECT (and may be in terms of hemodynamics), it would be important to know about CTp (if present); moreover, a propensity score matching might be more appropriate for the analysis;
4. it sound misleading to write in the abstract and text "numerically lower...", when the differnce is not statisctically significant;
The paper should be written taking into account the real significant differnces, and may be discussed later the difference in "trends".
Author Response
Please see the attachment.
Dear Dr. …,
Thank you very much for considering our manuscript for possible publication in Journal of Clinical Medicine. Here, we want to submit a manuscript version that has been revised with reference to the reviewers’ valuable comments.
Please find below a detailed point-to-point reply and a description of changes made regarding the given comments. Throughout the manuscript, revised sentences are marked in blue font.
Please do not hesitate to contact us for any further questions or considerations.
Yours sincerely,
Johann Pelz and Dominik Michalski
Reviewer 3
- It would be important to know the "vessel status" (in terms of occluded vessel and collateral score) in the two groups.
- The higher ASPECT in EVT can be the explanation of better prognosis at 3 months.
Reply: We would like to address both comments together, as, in our mind, they address the same issue. Instead of addressing the “vessel status” we measured the volume and proportion of cerebral edema and infarction on the last CT scan before decompressive surgery. Thus, even although both groups (endovascular treatment with decompressive surgery and only decompressive surgery) might have differed in terms of the localization of the occluded vessel and the collateral state, and, as proved, differed to the extend of reperfusion state (the reperfusion state (eTICI) is shown in Table 1, with probably no relevant reperfusion in the group without prior endovascular treatment), the proportion of cerebral edema and infarction referring to the whole brain volume was identical before decompressive surgery. Moreover, because of the same reason the higher ASPECTS in the initial cerebral CT is unlikely to explain the trend towards a better prognosis after 3 months. Even when focusing on the patients who presented with an ASPECTS <6, the proportion of edema/infarction was again similar.
There is the hypothesis of a differing speed of the growth of the ischemic core after large vessel occlusion, the so-called “late window paradox” which is probably due to a favorable collateral circulation (Albers GW. Late Window Paradox. Stroke. 2018 Mar;49(3):768-771. doi: 10.1161/STROKEAHA.117.020200.). On the other hand, there are patients with a poor collateral status who will have a large ischemic core despite an initial high ASPECTS and a swift and successful revascularization.
We added this to the limitation section: However, even if patients differed regarding the initial ASPECTS or the reperfusion state, the proportion of the cerebral edema and infarction referring to the whole brain volume was identical on the last cerebral CT before decompressive surgery, alleviating the importance of these parameters in the here investigated setting. As our study included only patients with DHC, the initial ASPECTS can thus be seen as a snapshot, while the growth of the ischemic lesion with an ultimately space-occupying effect depends on a variety of conditions, e.g., the collateral state. Consequently, some patients with LVO typically provide a slow evolution of the ischemic core and would thus benefit from EVT in an extended time window, also called the ´late window paradox´ [31]. These patients would probably not develop a space-occupying cerebral edema and, thus, were not examined in this study.
- As the two groups differs terms of ASPECTS (and may be in terms of hemodynamics), it would be important to know about CTp (if present); moreover, a propensity score matching might be more appropriate for the analysis.
Reply: For a propensity score matching one would need large cohorts of patients due to the various factors that might be considered. However, the volume of the ischemic core / edema before surgery was identical between both groups.
- It sound misleading to write in the abstract and text "numerically lower...", when the difference is not statisctically significant.
Reply: We focused on statistical significant differences between groups and removed the passages with “numerically lower …” values.
- The paper should be written taking into account the real significant differnces, and may be discussed later the difference in "trends".
Reply: The discussion was rewritten taken this comment in account.
Reviewer 4 Report
Comments and Suggestions for Authors
This retrospective study aims to investigate the functional outcome of ischemic stroke patients who underwent decompressive surgery depending on whether they had thrombectomy prior to surgery. For that purpose, the authors evaluated hospital records of 65 patients who had decompressive surgery. Their main finding was that no significant difference exists in the score on the modified Rankin scale after 3 months. However, there was a non-significant trend towards a better outcome in those who received thrombectomy, which was mainly driven by those who had a better ASPECTS score on admission.
Overall, this manuscript addresses an interesting topic. I have only a few comments:
#1. From the introduction, it remains unclear whether data from clinical trials or at least from case series exist that have shown a poor outcome in patients who underwent both, thrombectomy and decompressive surgery, or whether the hypothesis of a worse outcome is only based on theoretical considerations. If such studies or case series exist, they need to be referenced.
The statement in the introduction ‘In contrast to these concerns primarily based on observations away from modern reperfusion therapies, clinical data on the functional outcome after sequential endovascular treatment and hemicraniectomy are rare.’ needs a reference.
Reference 21 is more in favor of a similar outcome in those with and without prior thrombectomy.
The authors should consider also results of the following report:
Walter J, Alhalabi OT, Schönenberger S, Ringleb P, Vollherbst DF, Möhlenbruch M, Unterberg A, Neumann JO. Prior Thrombectomy Does Not Affect the Surgical Complication Rate of Decompressive Hemicraniectomy in Patients with Malignant Ischemic Stroke. Neurocrit Care. 2023 Aug 28. doi: 10.1007/s12028-023-01820-3.
#2. Results: Was there a difference in the time from symptom onset to hospital admission between patients who received thrombectomy and those who did not received thrombectomy?
#3. Overall, the discussion is too long and a bit clumsy. There are redundancies with the introduction and the methods section and also redundancies between different paragraphs of the discussion
#4. Discussion.
The sentence ‘While investigating the functional outcome in patients requiring decompressive hemicraniectomy due to space-occupying brain infarction, this study asked for a potential harmful effect of a priorly performed endovascular treatment, which has been discussed in terms of a reperfusion injury due to abrupt vessel recanalization adversely affecting surgical treatment.‘ needs a reference.
Author Response
Please see the attachment.
Dear Dr. …,
Thank you very much for considering our manuscript for possible publication in Journal of Clinical Medicine. Here, we want to submit a manuscript version that has been revised with reference to the reviewers’ valuable comments.
Please find below a detailed point-to-point reply and a description of changes made regarding the given comments. Throughout the manuscript, revised sentences are marked in blue font.
Please do not hesitate to contact us for any further questions or considerations.
Yours sincerely,
Johann Pelz and Dominik Michalski
Reviewer 4
- From the introduction, it remains unclear whether data from clinical trials or at least from case series exist that have shown a poor outcome in patients who underwent both, thrombectomy and decompressive surgery, or whether the hypothesis of a worse outcome is only based on theoretical considerations. If such studies or case series exist, they need to be referenced.
The statement in the introduction ‘In contrast to these concerns primarily based on observations away from modern reperfusion therapies, clinical data on the functional outcome after sequential endovascular treatment and hemicraniectomy are rare.’ needs a reference.
Reply: We thank the reviewer for his comment and rewrote large parts of the introduction, focusing on the known bleeding complications after decompressive hemicraniectomy by adding the systematic review of Kurland et al. (Complications Associated with Decompressive Craniectomy: A Systematic Review. Neurocrit Care. 2015;23:292-304. doi: 10.1007/s12028-015-0144-7).
- Results: Was there a difference in the time from symptom onset to hospital admission between patients who received thrombectomy and those who did not received thrombectomy?
Reply: We did not assess the time from symptom onset to hospital admission but from symptom onset to the first CT. This interval was dichotomized into ≤ 8 hours and > 8 hours. Due to local standard operating procedures, endovascular therapy (EVT) was routinely performed in patients presenting within 8 hours after symptom onset, while in those presenting in the extended time window (later than 8 hours after symptom onset), CT perfusion and ASPECTS were also considered in decision making.
EVT was performed more often in patients presenting within 8 hours from symptom onset than in those presenting later than 8 hours (34/39 versus 13/26, chi-square test p = 0.001). However, we did not add this data to the manuscript, since the focus was on the sequence EVT plus decompressive hemicraniectomy.
- Overall, the discussion is too long and a bit clumsy. There are redundancies with the introduction and the methods section and also redundancies between different paragraphs of the discussion.
Reply: The discussion was considerably shortened and relevant studies in the meanwhile (Walter et al., 2023) were included.
- The sentence ‘While investigating the functional outcome in patients requiring decompressive hemicraniectomy due to space-occupying brain infarction, this study asked for a potential harmful effect of a priorly performed endovascular treatment, which has been discussed in terms of a reperfusion injury due to abrupt vessel recanalization adversely affecting surgical treatment.‘ needs a reference.
Reply: We modified and shortened the conclusion section.
Round 2
Reviewer 1 Report
Comments and Suggestions for Authors
The manuscript has been improved sufficiently. No issues have been detected.
Reviewer 3 Report
Comments and Suggestions for Authors
All observation have been finalized